Double jeopardy: global change and interspecies competition threaten Siberian cranes

Gao Linqiang 1 gaolinqiang0@gmail.com
Mi Chunrong 1 2
1 Institute of Zoology, Chinese Academy of Science , Beijing , China
2 Princeton School of Public and International Affairs, Princeton University , Princeton, New Jercey , United States
Edwards Scott
Electronic publication date: 2024 Feb 28
Publication date: 2024
Volume: 12
Electronic Location ID: e17029
Received 2023 Oct 31; Accepted 2024 Feb 7
Copyright: © 2024 Gao and Mi
Copyright year: 2024
Copyright holder: Gao and Mi
License: This is an open access article distributed under the terms of the Creative Commons Attribution License, which permits unrestricted use, distribution, reproduction and adaptation in any medium and for any purpose provided that it is properly attributed. For attribution, the original author(s), title, publication source (PeerJ) and either DOI or URL of the article must be cited.
License URL: https://creativecommons.org/licenses/by/4.0/

Keywords: Siberian crane (Leucogeranus leucogeranus), Sandhill crane (Grus canadensis), Global change, Interspecies competition, IUCN, Arctic

Funding: Young Scientists Fund of the National Natural Science Foundation of China 32300420 This study is supported by the Young Scientists Fund of the National Natural Science Foundation of China (Grant No. 32300420). The funders had no role in study design, data collection and analysis, decision to publish, or preparation of the manuscript.

==============================
Anthropogenic global change is precipitating a worldwide biodiversity crisis, with myriad species teetering on the brink of extinction. The Arctic, a fragile ecosystem already on the frontline of global change, bears witness to rapid ecological transformations catalyzed by escalating temperatures. In this context, we explore the ramifications of global change and interspecies competition on two arctic crane species: the critically endangered Siberian crane (Leucogeranus leucogeranus) and the non-threatened sandhill crane (Grus canadensis). How might global climate and landcover changes affect the range dynamics of Siberian cranes and sandhill cranes in the Arctic, potentially leading to increased competition and posing a greater threat to the critically endangered Siberian cranes? To answer these questions, we integrated ensemble species distribution models (SDMs) to predict breeding distributions, considering both abiotic and biotic factors. Our results reveal a profound divergence in how global change impacts these crane species. Siberian cranes are poised to lose a significant portion of their habitats, while sandhill cranes are projected to experience substantial range expansion. Furthermore, we identify a growing overlap in breeding areas, intensifying interspecies competition, which may imperil the Siberian crane. Notably, we found the Anzhu Islands may become a Siberian crane refuge under global change, but competition with Sandhill Cranes underscores the need for enhanced conservation management. Our study underscores the urgency of considering species responses to global changes and interspecies dynamics in risk assessments and conservation management. As anthropogenic pressures continue to mount, such considerations are crucial for the preservation of endangered species in the face of impending global challenges.

Introduction

Anthropogenic global change is currently causing widespread biodiversity loss (Barnosky et al., 2011), placing thousands of species on the brink of extinction (Pimm et al., 2014; Ceballos et al., 2015). The Arctic, a unique and delicate ecosystem, harbors a diverse array of wildlife, yet it stands at the forefront of global environmental change, notably climate change (Gilg et al., 2012). Recent years have seen a pronounced impact on this region, marked by temperatures rising at a rate three times higher than the global average over the past five decades (Parry et al., 2007; AMAP, 2021). This rapid and profound ecological transformation (Box et al., 2019), has led to the shrinking of sea ice, glacier melt, and permafrost thaw (Davidson & Ruhs, 2021), consequently reshaping the Arctic landscape and habitats (Gilg et al., 2012; CAFF, 2013; Kubelka et al., 2022), Furthermore, these environmental shifts are significantly altering species distributions, interspecific interactions, community structure, and overall diversity (Harley, 2011).

The Arctic serves as the breeding grounds for only two crane species, the Siberian crane (Leucogeranus leucogeranus) and the sandhill crane (Grus canadensis). The Siberian crane, a critically endangered (CR) species according to the International Union for Conservation of Nature (IUCN, 2021), currently boasts a population of approximately 5,616 individuals (Wen et al., 2023). These cranes predominantly winter at Poyang Lake (Zhou, Ding & Wang, 1981; Shan et al., 2012) with recent sightings of wintering populations in the Yellow River Delta Nature Reserve, China (Huang et al., 2018; Wen et al., 2023). Their survival is threatened by habitat degradation, loss, human interference, and hunting (Meine & Archibald, 1996).

In contrast, the sandhill crane is classified as least concern (LC) by the IUCN and boasts the largest population among the fifteen crane species, estimated at 450,000 to 550,000 individuals, with a population on the rise (Harris & Mirande, 2013; IUCN, 2021). These cranes have an extensive distribution across North America, with a historical breeding range encompassing a small population in Chukotka, the northeastern corner of Siberia (Walkinshaw, 1949; Gao, Mi & Guo, 2019). Since the turn of the twenty-first century, the breeding population in this region has experienced growth, leading to an expanded distribution (Johnsgard, 1983; Bysykatova, 2012, 2013).

As the sandhill crane population expanded in the northeastern Siberia, their breeding areas began to overlap with those of the indigenous Arctic crane species, the Siberian crane. Observations indicate that when sandhill cranes encroach upon the Siberian crane’s range, they are met with aggression from the latter (Sleptcov & Vladimirtseva, 2019). Notably, this incursion of sandhill cranes has intensified in recent years, as they increasingly venture into Siberian crane territories for foraging, often resulting in confrontations (Sleptcov & Vladimirtseva, 2019). That suggest the emergence of a competitive relationship between these two crane species (Watanabe, 2006; Rozenfeld et al., 2023). This situation may exacerbate further if sandhill crane distributions continue to expand, whether due to population growth or global change, which in turn could impact the population growth of the critically endangered Siberian cranes.

Species distribution models (SDMs) have gained increasing importance in the fields of ecology, biogeography, evolution, and conservation (Guisan et al., 2013; Humphries & Huettmann, 2014; Mi et al., 2022) These models are widely employed to quantify species responses to global change, as demonstrated by their applications in studies on species distribution dynamics (Araújo et al., 2011; Newbold et al., 2020; Mi et al., 2023). SDMs play a pivotal role in elucidating how environmental changes have and will impact species ranges (Mi, Huettmann & Guo, 2016; Newbold et al., 2020). For instance, they have proven valuable in predicting shifts in the ranges of endangered species due to climate change (Velásquez-Tibatá, Salaman & Graham, 2013; Mi, Huettmann & Guo, 2016; Riquelme et al., 2018).

It is important to recognize that species distributions are influenced not only by abiotic factors, such as environmental variables, but also by biotic factors, including competitive relationships. Regrettably, many studies tend to neglect the impact of these biotic factors (Elith & Leathwick, 2009; Wisz et al., 2013). However, a handful of studies have demonstrated that incorporating interspecific relationships into SDMs can enhance the models’ predictive accuracy in estimating realistic species distributions (Tikhonov et al., 2017; Ye et al., 2022).

Rozenfeld et al. (2023) observed a westward shift of 400 km in the center of maximum sandhill crane density over a 28-year period. This shift coincides with a notable expansion in the overlapping areas of both crane species. Areas experiencing an increase in sandhill crane density are concurrently characterized by a corresponding decrease in Siberian crane density, with instances documented where the latter species has disappeared (Rozenfeld et al., 2023). How will global changes, encompassing both climate and landcover transformations, impact the range dynamics of Siberian cranes and sandhill cranes in the Arctic? Furthermore, is there a likelihood of intensified competition between these two crane species, potentially heightening the threat to the critically endangered Siberian cranes? To address these pressing questions, we employed ensemble species distribution models to forecast the breeding distributions of both crane species under the influence of global change. Additionally, we assessed the extent of range overlap to elucidate interspecies relationships, specifically the potential for increased competition. Our study’s findings hold valuable implications for enhancing the realism of risk assessments and the development of conservation strategies for this endangered species.

Materials and Methods

Occurrence records

We compiled a comprehensive dataset of occurrence records spanning from 1980 to 2023. These records included precise geographic information, such as longitude and latitude. We sourced this invaluable data from the Global Biodiversity Information Facility (GBIF; https://www.gbif.org/) for both Siberian cranes (L. leucogeranus, https://doi.org/10.15468/dl.8jtx8h) and sandhill cranes (G. canadensis, https://doi.org/10.15468/dl.zz7wq3). To augment our dataset, we also integrated post-1980 records for the endangered Siberian cranes obtained from the book ‘Threatened Birds of Asia.’ To ensure the reliability and quality of our dataset, we employed the ‘CoordinateCleaner’ package in R (Zizka et al., 2019), systematically filtering out records locate in capitals, institutes, and museums. Subsequently, we applied the ‘spThin’ package in R (Aiello-Lammens et al., 2015) to mitigate sampling bias, retained only a single distribution point within each square kilometer grid cell. Finally, 66 and 138 occurrence records were used for Siberian crane and sandhill crane, respectively (Supplemental 1).

Study area

We delineated our study area through the following processes: initially, we identified the geographical scope of sandhill crane breeding records in East Asia, as well as the breeding records of the eastern population of Siberian cranes (given that the middle and west populations of Siberian cranes are nearly extinct). Subsequently, we expanded this extent by 1,000 kilometers in the east, south, west, and north directions, encompassing the potential distribution range of these two crane species in the future, considering global changes. The selection of study area is based on cranes possible dispersal as indicated by prior research and model accuracy (Mi et al., 2017a; Warren, Matzke & Iglesias, 2020; Smith, Capinha & Kramer, 2022).

Environmental predictors

We obtained 19 climatic variables, representing current conditions (1970–2000) and future projections (2060–2080), at a 1 km resolution from the WorldClim database 1.4. We obtained landcover variable from (Chen, Li & Liu, 2022). To assess multicollinearity in our species distribution models (SDMs), we applied the ‘vif’ function from the ‘usdm’ R package to eliminate variables with Variance Inflation Factors (VIF) below 10 from the original set of 20 bioclimatic variables for each species (Naimi & Araújo, 2016; Mi et al., 2023).

Climate model projections

In consideration of the uncertainty inherent in future climate projections, we examined three global circulation models (GCMs): BCC-CSM1-1, CNRM-CM5, and MIROC-ESM (Mi et al., 2023). We assessed four representative concentration pathways (RCPs)–26, 45, 60, and 85–as representing future climate conditions during 2060–2080 (2070). These scenarios were chosen for their coverage of a wide range of plausible future global change outcomes and are widely employed in climate model projections (Borzée et al., 2019; Carlson et al., 2022). To mitigate uncertainties arising from variations in modeling techniques, we averaged the climate data across the three GCMs for each grid (Hole et al., 2009; Mi, Huettmann & Guo, 2016; Mi et al., 2023). Subsequently, we projected all climate and land cover layers to the North Pole Lambert Azimuthal Equal Area projection at a grid resolution of 1 km × 1 km.

Species distribution models

We employed the ‘sdm’ package (Naimi & Araújo, 2016) in R to implement an ensemble species distribution model for predicting species distributions. Our input data consisted of pseudo-absence records generated using the ‘gRandom’ method (Naimi & Araújo, 2016). We used 1,000 pseudo-absence records for each crane species. These pseudo-absences were derived within a calibration area surrounding species occurrence records (presence points) and delimited by a 100 km buffer around these presence records (Barbet-Massin et al., 2012; de Andrade, Velazco & Júnior, 2020).

To train our models, we utilized a random 70% sample of the initial data (comprising presence and absence records) and evaluated them against the remaining 30% of the samples. This process involved five iterations of split sampling to address data partition uncertainty (Thuiller, 2003). In summary, we employed four widely recognized SDM algorithms known for their high model performance in ensemble models: generalized boosted regression models (Graham et al., 2008), maximum entropy (Elith et al., 2006), random forest (Mi et al., 2017b) and support vector machines (Drake, Randin & Guisan, 2006).

To assess model performance, we utilized the area under the receiver operating characteristic curve (AUC) (McPherson, Jetz & Rogers, 2004), and the true skill statistics (TSS) (Allouche, Tsoar & Kadmon, 2006). We retained only those models with TSS ≥ 0.7 (Gallardo et al., 2017; Wang et al., 2017) and then constructed an ensemble model by weighting individual models based on their TSS performance for each species (Thuiller et al., 2009; Gallardo et al., 2017; Wang et al., 2017). Habitat suitability maps were converted to binary distribution maps (presence/absence) using the threshold that maximized TSS, a widely adopted method for generating potential species distribution maps (Barbet-Massin et al., 2012; Mi, Huettmann & Guo, 2016).

Results

Model accuracy

For Siberian crane, our models demonstrated good performance with an AUC of 0.947 ± 0.045 and a TSS of 0.849 ± 0.069. Similarly, for sandhill crane, the models exhibited excellent performance, with an AUC of 0.982 ± 0.013 and a TSS of 0.910 ± 0.040 (Fig. 1).

Figure 1 Model performance for (A) Siberian crane (Leucogeranus leucogeranus) and (B) sandhill crane (Grus canadensis).

The points in the boxplots represent the AUC or TSS values for each model corresponding to two crane species.

Variable importance

Our analysis revealed that precipitation during the wettest month (bio13) emerged as the most crucial variable influencing the distribution of both Siberian crane and sandhill crane (Fig. 2). Following this, the second and third most important variables for Siberian crane were precipitation during the coldest quarter (bio19) and mean temperature during the driest quarter (bio9). For sandhill crane, these were mean temperature during the coldest quarter (bio11) and mean diurnal temperature range (bio2). Land cover variables were comparatively less influential, ranking as the second least important and the least important factors affecting the distribution of these two crane species.

Figure 2 Variable Importance in species distribution models for (A) Siberian crane (Leucogeranus leucogeranus) and (B) sandhill crane (Grus canadensis).

Size = original. bio1 = Annual Mean Temperature, bio2 = Mean Diurnal Range, bio3 = Isothermality, bio7 = Temperature Annual Range, bio8 = Mean Temperature of Wettest Quarter, bio9 = Mean Temperature of Driest Quarter, bio11 = Mean Temperature of Coldest Quarter, bio13 = Precipitation of Wettest Month, bio15 = Precipitation Seasonality, bio19 = Precipitation of Coldest Quarter, landcover = land cover. Siberian crane image source credit: Zhilin Wu. Sandhill crane image source credit: iNaturalist, christian_nunes (https://www.inaturalist.org/photos/117000514?, CC BY-NC).

Spatial distribution changes under global change

Currently, Siberian crane primarily inhabits the northeast coast of Russia. They are distributed across Chukot, ranging from its central to western coastal areas, and in Sakha, spanning from the eastern to central coastal regions. Additionally, they primarily occupy Wrangel Island, covering its central and western sections. Sandhill crane can be found in the eastern coastal areas of Chukot, especially in the border regions with Yakutia. They also have a presence in most parts of Wrangel Island, the northern part of Kamchatka, and the south coastal areas of the Magadan. Both species share suitable habitats in the region between 160°E and 180°E. Their current suitable distribution areas mainly overlap in the central to western coastal regions of Chukot, including the border areas with Yakutia, and most parts of the central and western regions of Wrangel Island.

Under the influence of global change, the habitat of Siberian crane is significantly reduced on the north coast of Russia, shifting northward (Fig. 3). Suitable distribution on the mainland and Wrangel Island has nearly disappeared, with Anzhu Islands in the north becoming their primary suitable distribution areas. Sandhill crane will experience habitat loss on the north coast of Russia, particularly in the coastal area near the junction of Yakutia and Chukot. However, they expand significantly within the original range in northeast Chukot, north Kamchatka, and along the coast of the Magadan Region. This expansion also includes Anzhu Islands. Notably. The Anzhu Islands become the central overlapping area for these two crane species, and these spatial characteristics remain consistent across all four global change scenarios.

Figure 3 Spatial distribution of Siberian crane (Leucogeranus leucogeranus) and sandhill crane (Grus canadensis) under current conditions and four global change scenarios (rcp26, rcp45, rcp60, and rcp85) during 2060–2080 (2070).

The brown area represents the distribution of Siberian cranes only, the blue area represents the distribution of sandhill cranes only, and the red area represents the overlapping distribution of both species. Map from the World Bank (https://datacatalog.worldbank.org/search/dataset/0038272/World-Bank-Official-Boundaries).

Area changes under global change

Our analysis reveals contrasting patterns in habitat change for the two crane species under global change (Table 1 and Fig. 4). Siberian crane is projected to lose a substantial portion of its habitats, ranging from 85.3% to 92.7%, depending on the greenhouse gas emission scenarios (from rcp26 to rcp85). In contrast, sandhill crane is expected to experience a substantial expansion in distribution, ranging from 50.5% to 100.7%. Currently, approximately 18.7% of Siberian cranes habitats overlap with those of the two crane species. This overlap is projected to more than double, reaching 39.3% to 65.0% under different global change scenarios. The extent of habitat loss for Siberian crane and habitat expansion for sandhill crane intensifies with the magnitude of global change. Likewise, the proportion of overlapping areas within the two species’ distributions relative to the overall suitable habitat area for Siberian crane increases with the magnitude of global change.

Table 1 Suitable habitat area under different scenarios for Siberian crane (Leucogeranus leucogeranus) and sandhill crane (Grus canadensis) and their overlap (km2).

	Current	rcp26	rcp45	rcp60	rcp85	
Siberian crane only	227,393	36,703	23,455	22,869	7,100	
Sandhill crane only	135,423	258,784	302,834	339,538	363,719	
Overlap	52,337	23,732	17,620	26,837	13,199	

Figure 4 Proportion of suitable habitat area under different scenarios compared to the current suitable habitat areas for Siberian crane (Leucogeranus leucogeranus, brown dotted line) and sandhill crane (Grus canadensis, blue dotted line).

And the percentage of the overlapping suitable habitat area for both species concerning the suitable habitat area of Siberian crane (red dotted line).

Discussion

In our study, we have uncovered a stark contrast in the projected impacts of global change on two Arctic crane species. The nonthreatened sandhill crane (G. canadensis) is expected to witness a significant expansion of its suitable habitats, while the critically endangered Siberian crane (L. leucogeranus) faces a grim future with the loss of most of its habitats. Moreover, a larger proportion of the Siberian crane’s distribution areas will overlap with those of the sandhill crane, intensifying the risk to the Siberian crane.

Our model found the most important variable affecting the breeding distribution of both crane species is the Precipitation of the Wettest Month (bio13). Regions with lower precipitation appear to be more suitable for their distribution (Supplemental 2). Siberian and sandhill cranes are wader, their breeding success is highly sensitive to hydrological conditions, while high precipitation regions often with nest flooding (Haverkamp et al., 2022), that will reduce breeding success. Experiencing increased snowfall, rising temperatures, warm spells, and augmented warm-season rainfall leads more flooding in Arctic (Kane et al., 2008; Bintanja & Andry, 2017; McCrystall et al., 2021), which may highly impact their breeding. Siberian cranes, with a preference for relatively humid areas, might be more affected.

With only approximately 5,600 individuals remaining (Wen et al., 2023) and a continuing decline (IUCN, 2021), the Siberian crane is on the brink of extinction. Our findings indicate that their habitats will face extensive loss due to global change, further exacerbated by the encroachment of sandhill cranes (see Figs. 3 and 4). Additionally, the Siberian crane is confronted with competition from other species such as the tundra swan (Cygnus columbianus) and the bean goose (Anser fabalis) for food (grass and seeds, e.g., Arctophila fulva) and breeding grounds (Sleptcov & Vladimirtseva, 2019). There is an overlap in the foraging behaviors of the two crane species. Siberian cranes, akin to sandhill cranes, engage in land hunting before the ice melt, while sandhill cranes transition to water hunting around mid-June when aquatic food sources become abundant in wet areas (Watanabe, 2006). Sandhill cranes typically choose nesting locations without snow cover (Watanabe, 2006). If global changes lead to earlier snowmelt, the low basins currently frequented by Siberian cranes could potentially transform into suitable nesting habitats for sandhill cranes. Predators like the wolverine (Gulo gulo) and the Arctic fox (Vulpes lagopus) pose a threat by damaging Siberian crane nests and consuming eggs, thereby diminishing their reproductive success (Sleptcov & Vladimirtseva, 2019). In the future, the considerable habitat loss for Siberian cranes, coupled with an increasing overlap in habitats with the other crane species, may intensify competition for nesting sites and food resources (Fig. 3). Furthermore, the significant decrease in Siberian crane habitats could instigate a corresponding reduction in population size, leading to a decline attributed to inbreeding (Charlesworth & Charlesworth, 1987). This decline may manifest as diminished survival rates and reproductive success, ultimately propelling the population into an extinction vortex (Wright, Tregenza & Hosken, 2007).

In the past, the Siberian crane encountered a multitude of threats outside its breeding grounds. In recent decades, numerous critical stopover sites (e.g., Songliao Plain, Yellow River Delta) have been lost due to human overexploitation, wetland degradation, reclamation, grazing, and poisoning (Jiang, 2016; Li, 2016) The habitat quality of wintering grounds (e.g., Poyang Lake) has deteriorated due to various factors, including dam construction, aquaculture, and sand excavation (Buranham et al., 2017; Li et al., 2020). Additionally, there has been an increase in bird mortality during migration (Klaassen et al., 2014).

Compared with the non-breeding areas, the breeding grounds were historically regarded as secure refuges for the species. However, in the future, the Siberian crane will face a “double jeopardy” in breeding ground, with a negative impact of global change and high competition with other species. Which will make the breeding grounds no longer be a relatively safe place for this threatened species. Notably, in the context of global change, the Anzhu Islands are poised to serve as a critical refuge for Siberian cranes. However, they will also encounter heightened competition with Sandhill Cranes in this area. Future efforts should focus on bolstering conservation management in this region to reduce the risk of Siberian crane extinction.

Since the early 1990s, the population and distribution of sandhill cranes in western Yakutia have been on the rise (Bysykatova, 2013). Over the decade spanning from 1984 to 1994, the sandhill crane population in Yakutia increased by 1.8 times (Degtyarev, 2008). From 1995 to 2009, the number of sandhill cranes in the tundra near the Indigilka River surged by an astounding 13.3 times (Bysykatova, 2012). Notably, between 1988 and 2008, the breeding distribution in Yakut shifted westward by approximately 200–250 km. This expansion in distribution is likely attributed to population growth, paralleling the pattern observed in the eastern population, G. canadensis tabida, in the United States. For instance, the total number of sandhill cranes in the eastern United States grew from 423 in 1965–1966 to 46,194 in 2012–2013, with their range shifting north-northwest (Lacy et al., 2015). Furthermore, it has been speculated that long-term climate and land-use changes played a role in altering their distribution area (Lacy et al., 2015). Our study provides confirmation that climate and landcover changes indeed exert a significant influence on their westward and northward expansion (Fig. 3).

Numerous studies conducted in the Arctic have demonstrated that the significant breeding expansion of sandhill cranes in northeastern Russia has already placed competitive pressure on Siberian cranes (Watanabe, 2006; Germogenov et al., 2015; Rozenfeld et al., 2023). Our research has further revealed that the expanding habitat of sandhill cranes under the influence of global change will intensify the competition between these two crane species (Fig. 4), further escalating the risk of extinction for the Siberian crane. Moreover, our previous study documented an increase in the distributions and population of sandhill cranes in East Asia during the nonbreeding season, where they are becoming regular wintering birds (Gao, Mi & Guo, 2019). During this period, sandhill cranes coexist with five other crane species in East Asia (Gao, Mi & Guo, 2019). Recognized as one of the most versatile species (Reinecke & Krapu, 1986) with similar foraging habits (grains, seeds, tubers) as other crane species, they are likely to share habitats and foraging areas with these species. Should sandhill cranes continue to expand their range due to population growth or global change, this could significantly impact crane species that have long inhabited Asia, particularly the threatened ones such as the Siberian crane, red-crowned crane (Grus japonensis), hooded crane (Grus monacha), and white-naped crane (Grus vipio).

Previous studies focused on species vulnerability have primarily relied on the historical and current status of species populations and distributions, often using frameworks like the Red List. However, these assessments have faced criticism for their limited ability to adequately account for the risks posed to species by forthcoming global changes (Thomas et al., 2004; Bomhard et al., 2005; Keith et al., 2014). The influence of global change on animal populations has been widely documented (Alford, Bradfield & Richards, 2007; McMenamin, Hadly & Wright, 2008). Our study further affirms that global change exerts a substantial impact on the range dynamics of two crane species (Fig. 4). In addition, the advent of global change can intensify interspecies competition (Carrete et al., 2010), which often has severe consequences for threatened species (Bardsley & Beebee, 2001). Threatened species, characterized by low abundance and limited ranges, are particularly vulnerable to the adverse effects of heightened competition. This increased competition can lead to food scarcity, loss of breeding and wintering areas, and consequent reductions in reproduction and range size (Griffis & Jaeger, 1998; Hamel et al., 2013; Frei, Nocera & Fyles, 2015). Therefore, it is imperative that future species risk assessments and conservation planning take into account the responses of species to ongoing global changes and the potential impacts of interspecies competition (Hamel et al., 2013; Mi et al., 2023; Peng et al., 2023).

Conclusions

Our research has illuminated a critical phenomenon—the convergence of global change and interspecies competition, spelling a “double jeopardy” scenario for the critically endangered Siberian cranes, while simultaneously fueling the expansion of sandhill cranes in northeast Asia. This underscores the compelling need to incorporate species’ responses to environmental transformations and interspecies dynamics into the framework of species risk assessment and future conservation management. Furthermore, the increasing frequency of extreme flood events in the Arctic tundra poses a heightened threat to the reproductive success and long-term survival of cranes (Haverkamp et al., 2022). We propose the incorporation of climate extreme conditions in species distribution predictions. To safeguard the Siberian crane population from extinction, it is essential to undertake continuous monitoring and management. For example, driving sandhill cranes away from Siberian breeding areas to reduce competition, and constructing artificial nests for Siberian cranes to counter the effects of climate change or extreme weather, as studies have shown that such measures can enhance breeding success in other crane species (Cheng et al., 2022).

Supplemental Information

Supplemental Information 1 Occurrence records.

Supplemental Information 2 Response curves for for Siberian crane (Leucogeranus leucogeranus) and sandhill crane (Grus canadensis).

We thank Oleksandra Oskyrko for data collection of the Siberian crane in the book Threatened Birds of Asia. Special thanks to Zhilin Wu for providing the Siberian crane photo. We also thank two anonymous reviewers for their valuable and constructive feedback and suggestions.

Additional Information and Declarations

Competing Interests

Author Contributions

Data Availability

The authors declare that they have no competing interests.

Linqiang Gao conceived and designed the experiments, performed the experiments, analyzed the data, prepared figures and/or tables, authored or reviewed drafts of the article, and approved the final draft.

Chunrong Mi conceived and designed the experiments, performed the experiments, analyzed the data, prepared figures and/or tables, authored or reviewed drafts of the article, and approved the final draft.

The following information was supplied regarding data availability:

The raw data are available in the Supplemental Files.

We sourced this invaluable data from the Global Biodiversity Information Facility (GBIF; https://www.gbif.org/) for both Siberian cranes (L. leucogeranus, GBIF.org (12 October 2023) GBIF Occurrence Download https://doi.org/10.15468/dl.8jtx8h) and sandhill cranes (G. canadensis, GBIF.org (12 October 2023) GBIF Occurrence Download https://doi.org/10.15468/dl.zz7wq3). To augment our dataset, we also integrated post-1980 records for the endangered Siberian cranes obtained from the book ‘Threatened Birds of Asia’ (https://www.iucnredlist.org/resources/theatened-birds-asia).

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
