# Peer review of "Double jeopardy: global change and interspecies competition threaten Siberian cranes"

_PeerJ, doi:10.7717/peerj.17029_

## Round 0.1 · original submission · Major Revisions

The two reviewers both found the paper worthwhile but had some specific suggestions to improve it, including adding more detail on the justification and methodology of the study, additional annotations on some figures, and increased speculation on the implications of the ecological models for the physiology of the species. Also, the reviewers wanted you to comment on the effect of the study site and specific area on the SDM analysis. This I think is asking for a test of the robustness of your SDM given different sampling strategies for species points. You may have to conduct additional analyses to answer this question appropriately.

Reviewer 1 ·

Basic reporting

no comment

Experimental design

no comment

Validity of the findings

no comment

Additional comments

The manuscript investigates Double jeopardy: global change and interspecies competition threaten Siberian cranes (#92188).
The manuscript considers abiotic and biotic factors to predict breeding distributions through species distribution models (SDMs) and shows clear results. However, there are still some issues that need to be revised.I have some suggestions for improving the manuscript.
1.I suggest your discussion needs more details. I suggest you explain how important environmental variables affect the distribution of species and then analyze why are the important environmental factors that affect species distribution, For example, these environmental factors affect the metabolism of the species, or affect the food growth of the species, and therefore affect the distribution of the species.
2.I suggest that you improve the description at lines 245- 254 to provide more justification for your study. Is it possible to try to explain what specific food supplies are available in the study area for these species that have food competition, which creates food competition pressure.
3.I suggest that your conclusion needs more details. Is it possible to propose measures to improve the climate according to the species distribution predicted by the climate under different conditions, so as to help the species distribution.

Reviewer 2 ·

Basic reporting

The manuscript provides a comprehensive examination of the impact of global change and interspecies competition on two arctic crane species. The structure is generally clear, and the study is well-supported by data and literature. But the manuscript did not provide clear statement on the objective or hypothesis of the study. The authors should point out specific questions to address withi this research.

Experimental design

The authors should specify the algorithms chosen, how the models were validated, and any specific parameters or settings. The most important thing is the author did not tell the readers the location of the study area. For SDMs, the study area affect the results. If the author use different study area to predict the distribution, the author may get differnt results.

Validity of the findings

The study area could affect the results in predicting potential species distribution, but the author did not clearly specify the geographical scope of constructing the distribution model, so the results may not be robust.

Additional comments

In discussion, I just suggest the author to expand on the ecological implications of the observed changes. How might the projected loss of habitat for Siberian cranes impact their overall survival and breeding success?
For the figuers,more detailed annotations are needed, such as the meaning of the points in Figure 1 and the meaning of bio13 in Figure 2. It is recommended not to use both red and green in Figure 3 at the same time.

---

## Round 0.2 · Minor Revisions

It looks like reviewer 2 asks for a few additional minor revisions. Please respond to them in the manuscript and summarize your responses.

Please also consider thanking the two anonymous reviewers in the acknowledgments for their helpful comments.

Reviewer 1 ·

Basic reporting

no comment

Experimental design

no comment

Validity of the findings

no comment

Additional comments

Authors have attended my comments. I recommend this article to be accepted.

Reviewer 2 ·

Basic reporting

This is my second time reviewing the manuscript titled "Double Jeopardy: Global Change and Interspecies Competition Threaten Siberian Cranes." The manuscript addresses the habitat changes of two crane species in response to climate change, offering constructive ideas for the conservation of threatened species. However, the language in the manuscript requires refinement. My primary concern pertains to the selection of the study area—why did the authors extend it by 1000 kilometers around the nesting records?

There are also minor concerns:

Fig. 2: The authors should provide explanations for variables in the legend, such as bio 2, bio 1, and others.

Fig. 3: I recommend that the authors mark the time of rcp26, rcp45, rcp60, and rcp85 models.

Fig. S2: The author should include more detailed information in the legend. Additionally, enhancing the clarity of the figures is advisable.

Table 1: I suggest that the authors mark the units in the table for better clarity.

Experimental design

The author needs to clarify the scientific rationale for selecting the study area.

Validity of the findings

The findings are good

---

## Round 0.3 · accepted · Accept

Thank you for addressing the comments of reviewer 2 on your revision. I am happy to recommend acceptance of the manuscript at this time.